# Towards personalized care: Factors associated with the quality of life of residents with dementia in Australian rural aged care homes

**Mohammad Hamiduzzaman**[1]*, **Abraham Kuot**[1], **Jennene Greenhill**[1], **Edward Strivens**[2], **Vivian Isaac**[1]

1 College of Medicine & Public Health, Flinders University Rural Health SA, Flinders University, Adelaide, South Australia, Australia, 2 Cairns and Hinterland Hospital and Health Service, Cairns, Queensland, Australia

* mohammad.hamiduzzaman@flinders.edu.au

**Data Availability Statement:** Data cannot be shared publicly because of the terms and conditions contained within the ethics permissions granted for this study from Southern Adelaide

## Abstract

Quality of dementia care improves with a personalized approach to aged care, and knowledge of the disease process and unique care needs of residents with dementia. A personalized model of care can have a significant impact on the overall organizational culture in aged care homes. However, the dimensions of personalized aged care relating to dementia often remain under-managed. We aim to explore the factors that shape the dimensions of personalized dementia care in rural nursing homes using qualitative data of a mixed-method 'Harmony in the Bush' dementia study. The study participants included clinical managers, registered nurses, enrolled nurses and care workers from five rural aged care homes in Queensland and South Australia. One hundred and four staff participated in 65 semi-structured interviews and 20 focus groups at three phases: post-intervention, one-month follow-up and three-months follow-up. A multidimensional model of nursing home care quality developed by Rantz et al. (1998) was used in data coding and analysis of the factors. Three key themes including seven dimensions emerged from the findings: resident and family [resident and family centeredness, and assessment and care planning]; staff [staff education and training, staff-resident interaction and work-life balance]; and organization [leadership and organizational culture, and physical environment and safety]. A lack of consideration of family members views by management and staff, together with poorly integrated, holistic care plan, limited resources and absence of ongoing education for staff, resulted in an ineffective implementation of personalized dementia care. Understanding the dimensions and associated factors may assist in interpreting the multidimensional aspects of personalized approach in dementia care. Staff training on person-centered approach, assessment and plan, and building relationships among and between staff and residents are essential to improve the quality of care residents receive.

Clinical Research Ethics Committee, Australia and consented by participants. Focus groups and interviews were confidential to enable freedom of expression by participants, and participants consented into this study with an understanding that only anonymised quotes would be publicly available; not the entire transcripts as they contain health and well-being related personal data of the residents with dementia living in aged care facilities. Therefore, only relevant extracts from the transcripts, which qualify as the minimal data set, are included in this manuscript.

**Funding:** All authors received the following grant to conduct Harmony in the Bush Project: Research was financed by the Department of Health, Australia under Dementia Research and Innovation Grant project number 4-4ZOHI8C. MH is one of the Chief Investigators.

**Competing interests:** The authors have declared that no competing interests exist.

# Introduction

Personalized care is integral to the quality of residential dementia care [1, 2]. Dementia, a clinical syndrome that is progressive in nature, affects a person's intellectual capacity, communication, behavior and daily living activities, and the person with dementia experiences a life of disability and personality change [3, 4]. In aged care homes, standard dementia care is generally provided by a multidisciplinary team of nurses, allied health and care support workers, and related to the routine clinical assessment and maintenance of a holistic care plan [5]. The literature indicates a personalized approach, based on active listening, recognition, compassion, attentiveness and sensitivity to the resident's needs and preferences in providing care, promotes quality of life for residents with dementia (RwD) [1, 6]. Several studies investigate the factors and issues of personalized aged care or nursing home care such as, Rantz et al. (1998), Baldwin et al. (2013), Edvardsson, Sandman and Borell (2014) and Dyer e al. (2018); but the dimensions of personalized dementia care remain under-assessed. A lack of understanding of the dimensions of personalized aged care relating to dementia restricts the integration of person-centered approach into dementia care assessment and plan, especially in rural aged care homes [7, 8].

In Australia, there are approximately 425,416 people with dementia, with an estimated growth of 250 people each day [9, 10]. Of those, two in every five people live in regional towns and rural and remote areas, which places a substantial burden on the local aged care homes [11]. A total of 2,672 residential care homes are in operation in this country and 14% of these facilities are in outer regional, rural and remotes areas [11]. Residents with dementia account for 52% of all residents and about 73% of the total number of aged care staff are involved in providing direct aged care [10]. Research on rural residential dementia care reports a lower standard of care, and shows a high prevalence of falls, behaviors, comorbidities and depressive symptoms among RwD [12, 13]. For example, when a RwD experience falls and/or show agitation, these episodes tend to persist for a long time and sometimes remain ignored by the care staff, leading to reduced recognition and delayed management of co-morbidities. This results in significant psychosocial and work-related stress for care staff [14]. Adopting a personalized model of dementia care lifestyles and activities may have beneficial influence on both residents and caregiving staff.

The recent renewal of the Aged Care Quality Principles of Australia set out new quality standards for aged care homes in regard to clients' dignity, care planning with consumers, individualized services and supports, organizational environment and governance [15]. As such, the provision of personalized dementia care in rural aged care homes has been receiving increasing attention, including how to plan and design care management, and how the quality of care is linked to the organizational characteristics [12, 13]. The focus of the empirical literature on the evaluation of personalized care depends on clinical outcomes, which has been considered as an indicator for ensuring quality in care [16, 17]. This attention to the clinical outcomes contributes to a lack of focus on the operational aspects of personalized care management that are of value to the RwD [17]. For examples, lack of resources and support from organizations and colleagues have negative consequences on the care staffs' services. Current literature also relates the quality of personalized care to the knowledge and skills of staff such as a lack of understanding of dementia, stress and reluctance [12]. According to the Dementia Australia National (2019), it is unknown how the quality is maintained in dementia care and whether the staff value the aspects of personalized dementia care. We aim to provide an insight into the dimensions including factors and issues that influence personalized dementia care in rural aged care homes, using qualitative data from 'Harmony in the Bush' project.

## Materials and methods

This qualitative study is a part of a large project, 'Harmony in the Bush' (2017–2019)–funded by the Australian Department of Health, that seeks to understand the personalized care support for RwD in Australian rural aged care homes and trials a personalized dementia care program.

### Theoretical background

We used the principles and dimensions of a multidimensional theoretical model developed by Rantz et al. (1998) in this paper to provide a theoretical structure of the study findings [18]. This model was emerged from an inductive analysis, and useful in the identification of the complex nature of care quality in aged care homes. There are seven dimensions in the model including *central focus*, *interactions*, *milieu*, *environment*, *care*, *staff* and *safety* [18]. Each of the dimensions provides directions for quality measures [Table 1]. As a result, we applied them in the analysis and discussion sections to explain the factors impacting on quality of personalized dementia care, and to report any new dimension or determinant (if any) to the model.

### Data collection

Following the ethics approval from the Southern Adelaide Clinical Human Research Ethics Committee (*Project Number*: *277.17*), we approached staff working in five rural aged care homes of South Australia and Queensland to participate in the project; by contacting managers of each site. These aged care homes included two privately owned, one public funded, one Aboriginal, and one not-for-profit center. An Information sheet and consent form were emailed to the facility managers for distribution among staff. Participants were free to contact the investigators if they wished to be interviewed and or to participate in focus group. They were also informed that they could choose to withdraw at any time of the project without adverse effects on their immediate or future care provision. The three inclusion criteria in the recruitment of staff were: (i) working in aged care center; (ii) have experience in caring RwD; and (iii) agree to provide consent and participate in a face-to-face or telephone interview and or focus groups. We informed the staff that pseudonyms were to be used in reporting the findings to maintain anonymity and brevity.

One hundred and four care staff participated in interviews and or focus groups at three phases: post-intervention [interview: 38; focus group: 13], one-month follow-up [interview: 11; focus group: 2] and three-months follow-up [interview: 16; focus group: 5]. Sixty-five audio-recorded semi-structured interviews and 20 focus groups were conducted. The interviews started with open-ended questions, which were followed by prompts to guide the discussion and covered the topics including: their participation in the project, understanding the needs of RwD, care management, shared values and relationships among staff, job stress and

**Table 1. Dimensions of multidimensional nursing home care model.**

| Dimensions | Operational areas |
| --- | --- |
| Central focus | Acknowledge the needs of residents and importance of their family members perspectives |
| Interactions | Communication between residents and staff |
| Milieu | Overall situation and sensations of residential aged care homes such as calm and friendly place |
| Environment | Aged care centers consist of various components such as room, furniture and decoration |
| Care | Staff should have good understanding about each residents' healthcare and wellbeing needs |
| Staff | Knowledge and skills of staff |
| Safety | Residents must feel safe, secure and free to talk with staff and obtain support when they are in need |

well-being and organizational environment and support. In this study, data saturation was achieved in 45 interviews and 14 focus groups as the answers to the questions became repetitive at these points. However, the ongoing interviews and focus groups at different time-points continued to explore new insights. Audio recordings of focus groups and interviews were transcribed verbatim by a professional research data transcription organization.

## Data analysis

Two researchers were involved in the data analysis process and each of them started with independently importing the data into a qualitative data analysis software NVivo 12. We used the thematic analysis technique of Braun and Clarke (2006, 2014) in coding and analyzing the data that included several steps [19, 20]. An initial process of listening to the audios and reading the transcripts was undertaken to be familiarized with the data. The second step was open coding that provided an insight of each phenomena related to the personalized dementia care. A 'topic coding' was conducted as the third step to search the patterns of codes across interview/focus groups' questions. The fourth step involved a categorization of codes into nodes or candidate sub-themes. In next step, we reviewed the candidate sub-themes, based on the context and generated candidate themes. Finally, sub-themes and themes were defined and presented. In order to maintain reliability and validity of the findings, the themes emerged from each of the researcher's coding and categories were discussed and reviewed in the 'Harmony in the Bush' project's weekly meetings [21]. This group discussion was also about defining, naming and contextualizing the themes to represent the contextual and practical realities about how the factors impacted on the staffs' services and quality of life RwD.

## Findings

### Baseline demographics

The demographic information of the staff participants is presented in Table 2. A total number of 104 staff participated, and of them 84% were female. The mean age of the staff was 45 (ranged from 19 to 72 years). Most of the staff (43%) completed Certificate III in ageing or

**Table 2. Demographic characteristics of all participants.**

| Characteristics | Staff (n = 104) |
|---|---|
| *Gender (%)* | |
| Female | 87 (84) |
| Male | 17 (16) |
| *Mean age (SD*)* | 45 (13) |
| *Highest Level of qualifications (%)* | |
| Certificate III (Higher Secondary) | 40 (43) |
| Enrolled Nurse (Undergraduate) | 15 (14) |
| Registered Nurse (Graduate) | 16 (15) |
| *Country of origin (%)* | |
| Born in Australia | 66 (63) |
| Born in overseas | 44 (37) |
| *Language spoken at home is not English (%)* | 31 (30) |
| *Carers work Full-time (%)* | 28 (27) |
| *Working in residential age care for less than 5 years (%)* | 45 (44) |

*Standard deviation

disability prior to joining residential aged care. It is important to note that 37% of the total staff were born overseas including Philippines (11%), India (8%) and New Zealand (5%); while 31% of them did not speak English at home. In addition, there were 73% part-time and casual staff and 44% had residential aged care work experience for less than five years.

## Dimensions of personalized dementia care

The major themes and sub-themes that emerged from the analysis of the qualitative data included: resident and family [resident and family centeredness, and assessment and care planning]; staff [staff education and training, staff-resident interaction, and work-life balance]; and organization [leadership and organizational culture, and physical environment and safety]. The explanation of the themes including the excerpts of the participants are presented in the following sections.

**Resident and family.** The first main theme presented the staff perspectives of the provision of care to RwD in rural nursing homes. Dementia care was related to a lack of recognition of each resident's care needs and exclusion of family members' perspectives. Assessment and care planning meant a generalized approach for the residents.

*Resident and family centeredness*. The sub-theme of 'resident and family centeredness' represented the status of RwD and their family members in the rural aged care homes relating to the recognition of the residents' needs, inclusion of the voice of residents' family members in care services; and facilitation of social interactions between residents and their family members.

Most of the care staff identified recognition of needs as an essential part of their daily focus in providing care to the RwD. This recognition was related by most of the participants to the awareness of each of the RwDs' agitation and falls, frequent communication and taking care of their shower, food and sleep needs. For example:

*At the end of the day, we are here for the patients. You know if they agitated if they are having more falls, things are happening that should not be happening, yeah that part of our duty of care. Yeah if they calmed that make us less stress.*

(Staff, ACH 1 –Post-intervention interview)

*Yeah, because if you don't, you know, you need that communication all the time, because we're here for the residents, not ourselves, so, you know . . .*

(Staff, ACH 5 –One-month follow-up focus group)

However, some participants clearly stated that the staff were dealing with the symptoms of RwD and doing their routine tasks without differentiating the needs between RwD and residents without dementia, as two staff stated in the following excerpts:

*As I said, they* [aged care workers] *think more about–not about the person but the person living with dementia. So, they try to deal with the symptoms not with the person. So, as I said, giving you the example. They* [aged care workers] *try to live in their world, not bringing a resident who's got dementia into a normal person's world.*

(Staff, ACH 4 –Post-intervention focus group)

*. . . you know, like things that we used to do that we individual for each person, that we weren't really aware that we were doing for them* [residents with dementia] *. . .*

(Staff, ACH 3 –Three-months follow-up interview)This lack of differentiating the choices of RwD from general aged care was closely related with the inclusion of family members' perspectives in residential aged care.

The study findings confirmed that there was some degree of exclusion of the family members' perspectives in providing care to the RwD. Some staff mentioned about "*allowing family members' visits*" at aged care homes and using *feedback forms, phone calls and formal gatherings* [i.e. Christmas party, NAIDOC Week] in understanding the needs of the RwD. However, the quantity and quality of interaction [e.g. information exchange, shared decision making] between staff and family members regarding the service and preference of these residents was described by the majority of participants as inadequate.

> *. . . we see their* [residents] *family members visit here and meet their parents on different occasions such as celebrating birthdays. Some families come in once or twice every month, some do not visit very often. We discuss with family members* [legal guardian] *for each resident's medical history at admission time, but I do not see continuous engagement of staff with residents' family members* [in providing care]
>
> (Staff, ACH 4 –One-month follow-up interview)
>
> *We used to have a, for example, we have just done last year Christmas party and NAIDOC week, things like that. . . . all the families were invited here. They come and they interacted with residents. They had free food and they get together. Its happen I can say twice a year . . . not on a regular basis because of the issue of staffing. . . . everyone one is too busy with their tasks. It used to be happened a lot, not its* [programs] *reduced I must say . . . so the family participation is very less.*
>
> (Staff, ACH 5 –Three-months follow-up focus group)
>
> *. . . you know, there's a book, ah, that was started for families to make comments, and I know that in that book some families have made comments in that book . . . not many.*
>
> (Staff, ACH 2 –One-month follow-up interview)

It was conveyed that the perspectives of family members were underrepresented in residential dementia care because it was not a priority for management and staff.

The regular social interaction of RwD, according to the staff, with their family members was compromised. Most of the staff described positive outcomes as a result of family members' visits at aged care homes relating to the mobility, interactions and food intake among RwD. For example,

> *For them, plus the way they interact with the family too, that's what I watch as well, you know, how they are with their family and when you've been nursing someone for a while, you see the change. The change is wonderful–they enjoy every moment with their family—listening to music together and laughing. I see a lady who does not talk much with staff, but she interacts very well with her family member.*
>
> (Staff, ACH 1 –Post-intervention focus group)

In contrast, family members' regular visits and meeting with RwD was found by some of the staff as limited because of a provision of permission and a lack of interest among family members or relatives.

*Yeah and yeah, the family have been very respectful of everything . . . they asked whether they are allowed to go in there or, you know, and they need sort of asking permission now.*

(Staff, ACH 2 –Post-intervention focus group)

*It* [interaction between family members and RWD] *hasn't really - - - We haven't really seen it. If family members are there then, yeah, they'll join in but it all depends on the time, like, if the family members are there or not.*

(Staff, ACH 4 –Three-months follow-up interview)

In general, staff did not place a major emphasis on family interaction. It was not seen as a core responsibility for all staff; rather the onus was on the family to make an effort to visit and the relationships between staff and family members depended on personal preferences and was patchy. Some family members were blamed if they made a complaint or were too demanding, whilst others were afforded respect and care.

*Assessment and care planning.* Assessment and care planning emerged as two fundamental aspects of personalized dementia care practice in the views of most staff. The participants described the care approach and plan in relation to the acceptance of daily care challenges and generalized care plan in place.

In describing the care management for RwD, most of the staff described the day-to-day care challenges as being "*everyday was different*" for them. Some staff who worked in dementia unit specified the challenges of dementia care such as residents' behavior, verbal aggression, wandering, lifestyle and personal needs, were important to include when assessing and providing holistic care to RwD. For example:

*People coming into the facility, I get the feeling or what I have seen over the time that people would not come when they need a less help. They come here when they need a lot more help– coming here with comorbidities, extreme cases of whether it is physical or mental issue . . . so we get these people now before we got to know them and you see these climbs and build the strategies as you go . . . so it hard to get adapt you with extreme behaviors such as screaming and wandering.*

(Staff, ACH 1 –Post-intervention interview)

*'I think every day is not perfect, but it's to be expected like some patients might get more sleep and might a lot of rest in the afternoon . . . sometimes it does not work, for example, one evening two residents were close to each other and the male resident says some nasty things to female resident, and then she gets agitated. This night was not good at all for the carers. It was frustrating . . .*

(Staff, ACH 4 –Post-intervention interview)

*Some of them* [Staff] *have taken it* [needs of residents] *on board and, and they're terrific, but there are, there are a few that are still a bit task driven.*

(Staff, ACH 3 –Three-months follow-up focus group)

The statements indicated that the complexity of dementia care and differences in everyday dementia care service at the presence of task-driven work environment contributed to a generalized approach to the RwD.

Another aspect to include is the individualized care plan for RwD. There were the residents' needs and wishes to integrate them into care plan. It was found, in particular situations when

staff were involved in care for residents with advanced dementia, that most of the care staff explained their routine tasks related to dementia care. For example:

*I'm a nurse care assistant and I have the residents to provide the basic daily care, um, and also helping them improve their independence here and, yeah, and helping them for feeding, and that things.*

(Carer, ACH 2 –One-month follow-up interview)

*The role people are doing is very defined. It does not really reinforce anyway that there are opportunities, for example, this kind of music intervention what are trying to do. Our tasks for the residents include help them in showing, dress up and having meals or take them to the tv room for lifestyle activities.*

(Staff, ACH 3 –Post-intervention interview)

One staff clearly stated about the presence of a generalized approach for the RwD.

*I think that is the big thing I gathered from the bulk of people is that they may not have seen from an individualized approach . . . they are looking for a box everybody fits into it.*

(Staff, ACH 2 –One-month follow-up focus group)

In addition, several staff described the potential of individualized and creative dementia care plan. Most of the staff who participated in 'Harmony in the Bush' project considered the positive outcome of non-pharmacological personalized care interventions [e.g. reading bible with the RwD, no-wake up policy, personalized music] and found the positive behavioral change of RwD.

*For example, yeah, one of the residents, he did not want to go in a group activity, so when he did indicate, you know, one of the staff member go sit with him and liked to do, you know believe in God, so they read the Bible for him and they calm him down so that's a one on one activity.*

(Staff, ACH 4 –Post-intervention focus group)

*. . . one when we started the Harmony in the Bush, she would just spell, just spell the same word over and over and over and she'd come up to you and spell the same word. We went through a stage* [personalized music intervention] *where she would then, she started talking just in sentences, she would tell you what she wanted, if you initiated conversation with her. She would answer appropriately or whatever.*

(Staff, ACH 3 –Post-intervention interview)

Even though the use of care plans was evident in all aged care homes, the components of assessment and care plan did not adequately incorporate personal need and preferences of RwD, as indicated by the staff who participated in this study. This lack of individualized dementia care plans created barriers for the staff to ensure quality in dementia care, which was also related to the inclusion of personal assumptions rather than evidence-based care practice by the staff.

The study findings revealed a lack of evidence-based care plans for RwD, especially among the aged care staff who were involved in providing dementia care. It was found in the interviews that the care staff made some assumptions that are more generic in nature about the

RwD and their care needs [e.g. RwD cannot understand the living environment or lack of innovation in care plan].

*. . . there's also another resident has, um, opened up to me whereas, um, prior to this program happening I'd hardly get boo out of her, and whereas, um, . . . you know this is dementia and they live in dementia world. In the dementia world, you know they may be having really hard time in understanding the environment.*

(Staff, ACH 2 –One-month follow-up interview)

*I don't know. I don't think it's* [music] *more important than the sleeping in policy, but I think that all of the aspects* [of the project] *help with the mood and how the flow of the day goes and . . .*

(Staff, ACH 4 –Post-intervention focus group)

These assumptions were not evidence-based, and they had an influence on the staff's service that impacted on the quality of life of RwD, as stated by one staff in the following excerpt:

*. . . it is very hard for the staff to establish those intervention in place with those dementia residents, so that's the main reason, you know, understanding of the staff for the resident, you know, you can, you can sleep as long as you want, we're not going to wake you up, because automatically they wake themselves but they want to get out from bed. It's like dementia is the reason you know. Your understanding of the resident with dementia.*

(Staff, ACH 1 –Post-intervention focus group)

The generic assumptions among the aged care staff negatively influenced the residential dementia care.

This theme of resident and family has been shown to be a result of a reluctance among staff in recognizing the care needs and preferences of RwD and inclusion of their family members, and also an absence of person-centered dementia care service due to a lack of knowledge and skills of staff.

**Staff.** The second main theme 'staff' described a lack of access for staff to ongoing education and training opportunities, and this was related to an isolation of RwD in staff-resident communication. Difficulty in managing professional and personal life influenced the performance of staff.

*Staff education and training.* The sub-theme of staff education and training highlighted a lack of relevant education and inadequate on-going educational opportunity in contributing to a lack of understanding and knowledge about the person-centered dementia care in aged care homes.

Most of the care staff agreed that they had a lack of knowledge about dementia care and this lack of knowledge was a result of little or no prior education about dementia signs and symptoms, assessment process, preventable measures and principles of person-centered care. For example:

*. . . I write a literature review on how to manage pain in people with dementia. I speak to* [manager] *here because I feel that lot of the new registered staff here, they just new, they do not understand that when you complete the pain assessment, it needs to be done while person moving. We are getting a lot of false assessments coming through with no pain, no pain and no pain.*

(Staff, ACH 1 –Post-intervention interview)

*Yeah, we need more education for our staff as well for dementia. . . . We have totally fresh staff who don't know nothing about dementia. Just need education to everybody to keep it up. . . . Yeah, the training does help implement everything and it reinforces everything that you guys have said and taught.*

(Staff, ACH 5 –One-month follow-up focus group)

Some staff also stated how they learnt about the likes and dislikes of RwD, as described by one staff in the following excerpt about her prolonged engagement with one resident with dementia:

*There is one resident in there that, like I can understand her better from when I first engaged with her, I can now understand what she wants, like, you know, come on, we'll go for a walk, and yeah, she's talking and then, yeah, I can. I think*

(Staff, ACH 4 –Post-intervention interview)

A lack of education on dementia was evident among the staff that restricted their capacity to provide personalized dementia care to the residents when needed.

Most of the staff identified the on-going education and training sessions as essential part in providing care to the RwD. While thirty-five staff highlighted their participation in on-going information sessions, the rest of them disagreed that there was sufficient training available on dementia care. Two staff clearly stated about the requirement of more on-going education and educational intervention projects in order to improve personalized dementia care.

*'I think it needs more. I mean this is my opinion. We do staff training on dementia or challenging behaviors . . . but staff can't handle it because of that* [lack of ongoing training sessions]. *I think we need more education and intervention when we participate in project like this.*

(Staff, ACH 5 –Three-months follow-up interview)

*So we do start [to struggle with] dementia behavior and challenging behaviors and we just recently had a staff can't cope and had to go on stress leave because of this, so, yeah, I think the more education than we're doing, intervention when we're doing like projects like this. . . . It's a lot of stress we need more education and resources and training in dementia.*

(Staff, ACH 4 –One-month follow-up interview)

It was also explored in the focus group discussions that there was a need for a full-time trainer (or more trainers) in all aged care homes who could facilitate on-the-job training for staff about personalized dementia care.

*. . . if we have that one person [the trainer] always who is making sure that everything is happening* [appropriately] *on daily basis, morning, noon and night, that's would be incredible.*

(Staff, ACH 3 –Three-months follow-up focus group)

*I was a trainer, a workplace trainer, in my previous role. Education was part of my role. I do it opportunistically, I am doing it without people realizing on doing it and that's the same*

*with young registered staff with medications . . . I am the only one person doing it. I do, but that's not my role. I do not have capacity or time to do it.*

(Staff, ACH 1 –Post-intervention interview)

In addition to a lack of scope, there was no comprehensive staff training and development, and this was linked to a relegation of residents in regular communication with staff.

*Staff-resident interaction.* An isolation of RwD in patient-staff interaction related to a sense that there was not enough emphasis on the person-centered relationships and empathetic behaviors to the residents. A frustration was found in most of the participants' voice about staff's relationships with RwD, which was described in the light of the residents' behavioral and psychological changes and high rotation and time commitments of staff.

Most of the staff described how they managed these behaviors using personal interactions and empathy that resulted in less behaviors among the residents. However, it was found in the interviews and focus group discussions that some staff were not competent enough in accepting and engaging themselves in managing the challenging behaviors of the RwD, for example:

*Yeah, like you're trying to do something with them and they're not lashing out at you.*

(Staff, ACH 1 –Three-months follow-up interview)

*. . . whereas before, it took two people to get her on the shower chair, to get her in the shower. You didn't always get her there . . .*

(Staff, ACH 4 –One-month follow-up interview)

A difficulty of majority staff to be empathetic to the challenging behaviors of RwD led them to a stressful work experience that had an impact on the quality of interactions with the residents, which was also a result of high rotation of staff and time restrictions.

*I can see the behaviors* [of some residents] *that cross those boundaries. I haven't been very successful with handling that. . . . when I talk about staff, it must be extreme behaviors. They may not admit to it. But also noise, if you have resident who walks around and screaming all the time. . . . they may be aggressive, they may be violent, they may be verbally aggressive–that sorts of things.*

(Staff, ACH 2 –Post-intervention interview)

Two important aspects also emerged in the views of participants when they discussed about person-centered interaction between staff and RwD such as high rotation of staff and commitments to the routine tasks. According to some participants, there was a high rotation of staff in all aged care homes and this was found in the descriptive statistics (73% of the total number of staff worked part-time or casual). The impact of such high rotation on residential dementia care was distinctly described by one participant in the following excerpt:

*Ah, a bit more, like you say, there's, there's a high rotation of staff. I think maybe it would be better to have more regular staff, so they've got familiar faces, people know them better, um, yeah, when, when there's, ah, I mean I know when you meet new staff there's, there's a bit of getting to know you, but, um, and, and some of the staff that just don't respond well to the, the dementia wing; I don't think they should be rostered in there.*

(Staff, ACH 3 –Post-intervention focus group)

Some staff mentioned their commitment to the routine tasks and how time restriction impacted on their care support and interactions with the RwD, for example,

*Because then with that stress, if I think you've got to be finished like 11 o'clock, that's just too impossible, you know, like they're not pieces of meat, they're people. So, there's no emphasis on getting, this is the time you've got to them finished, you know. . . . it becomes hard under such pressure to remain stress-less and provide proper concentration for RwD.*

(Staff, ACH 2 –Post-intervention interview)

*we need to just work out the time management part of it. Sometimes it works and sometimes it's hell on earth, but we're still doing it, we're still initiating it.*

(Staff, ACH 5 –One-month follow-up focus group)

Because of high rotation of staff and time restrictions, the staff found it hard to manage the person-centered interaction with RwD that resulted in an ignorance of the residents' needs and preferences. This lack of care support for RwD was also identified by some staff in relation to their personal and social life at home and in society.

*Work-life balance of staff.* This sub-theme meant a living circumstance of the care staff where their care service occurred in a context of personal and family issues, social isolation and stigmatization that contributed to a reluctance in caring RwD.

While most of the staff stated that they successfully separated their family and work life, some carers described the impact of personal issues on their care support to the residents such as unwillingness to work and drugs and alcohol.

*I have been working for more than 13 years in Australia. When I had child [pregnant], I took maternity leave and had my time. But I can manage my work stress and home stress. I never bring [family issues) in to work and it never affect my availability or my work task at all.*

(Staff, ACH5 –Three-months follow-up focus group)

*There are so many issues why they are not coming to work. Not sure if it appropriate to comment on that. Its cultural issue from my understanding. Drug and alcohol are a big factor here in the community. These are the things that are affecting their work performance. They do not want to come to work because they do not think they need a job. They can sit home and access center-link money. When you can earn money sitting at home, why you will go for work.*

(Staff, ACH5 –Post-intervention interview)

One staff also discussed about how the impact of having young family and children impacted on the work performance.

*. . . but personally, when I get home, I get more easily stressed from my children and other stressors that are pushing you back at higher level. I personally talked with some people and they mentioned about it to me, typically people with young children.*

(Staff, ACH 4 –Post-intervention interview)

*. . . I think for younger people [carers], they may not have a partner or a family yet. I basically have a family since I work here. I have seen a lot of people stresses at home increase.*

(Staff, ACH 2 –One-month follow-up interview)

Such personal and family issues created difficulties for the staff in concentrating on the care support whereas RwD were in need of individualized care. These issues in combination with a social isolation and stigmatization resulted in a reluctance among the staff in providing care.

Social isolation and stigmatization were found by some staff as having negative impact on their mental well-being, as described by two staff in the following excerpts:

*. . . we may not react here, but [react] at home or in public because of the work obviously with these people reflected by the disease.*

(Staff, ACH 2 –Post-intervention interview)

*From the community in general, I feel like aged care is stigmatized something like it is a professional who are not able to do other works. And the status is like sitting in the background . . .*

(Staff, ACH 1 –Three-months follow-up focus group)

This staff furthered her discussion about how the aspect of social stigmatization influenced on the staff's focus that created a reluctance in care support to the residents.

*I think there are some carers who were reluctant and work necessarily having not an open mind that why we are giving them certain things. I think that's probably the biggest factor in providing care.*

(Staff, ACH 1 –Post-intervention interview)

Thus, according to the staff, the personal, family and social issues were important to consider in understanding their work performance for the RwD.

Some staff had access to ongoing education and training opportunities and others didn't, and the organizational environment and culture were often focused on rules that would provide a safe workplace rather than encouraging staff to think of ways to implement continuous quality improvements.

**Organization.** The third main theme 'organization' contextualized the role of leadership, staff relationships and physical environment of rural nursing homes in ensuring personalized care for RwD.

*Leadership and organizational culture.* Leadership and organizational culture were found by the staff as key to practice a holistic care management plan for RwD. The participants identified three aspects of relationships that had influence on their care service to the RwD including hierarchical authorization, an avoidance of building relationships, cultural diversity and generation gap in staff.

Some staff stipulated that how authoritative leadership influenced on their care activities. While some clinical managers discussed about the difficulties in engaging the care workers into personalized care, several care workers emphasized on the need of improving respect among the staff [horizontal and vertical] in order to implementing new model of care.

*But there will be 90% of staff will say, we're okay, we don't want to change things. So, changing staff approaches or changing staff minds was really challenging. . . . they all got on board and I think–now I don't have to tell them that, no, we're going to stick with it.*

(Staff, ACH 4 –Post-intervention interview)

*I think respect between staff* [clinical managers and aged care workers], *if that–am I–am I allowed to say that? . . . I think that's a big one, I think that if we improve that* [respect between and among clinicians and aged care workers] *then that also reflects back on them* [residents] *and that also shows them* [residents] *that we're respectful, everyone's respectful to everyone and everyone's. . .*

(Staff, ACH 2 –Three-months follow-up interview)

As a result, according to the participants, avoidance of building relationships with supervisor and colleagues was common in aged care homes. In reply to the question about the quality of personal interaction with other staff, some staff stated about their avoidance of building relationships and sharing new information to ignore potential threats from the management, as one staff stated in the following excerpt:

*Exactly, and then you don't want to be dobbed into the boss, you know, and get a phone call the next day saying, "Why did you have a go at this staff?" and it's just like "No," . . .*

(Staff, ACH 3 –One-month follow-up focus group)

In contrast, the staff working in two aged care homes described about the opportunity of buddy shifts that helped them to build relationships with other staff that led to a better understanding of the needs of RwD.

The aspect of diversity in care staff challenged the quality care service for the RwD. This diversity in staff was related to the cultural background and generation gap. Most of the participants agreed that a significant proportion of staff were from overseas, as identified in the demographic profile of staff participated in this study (37% of the total number of staff born overseas), who had different cultural backgrounds with different language and likes and dislikes that impacted the quality of care support to the RwD.

*We also have staff from different cultures and backgrounds, but a good chunk of carers is our first-generation migrants, and this is a challenge with language and culture . . . it could also be a drawback. . . . some of them are very different than we used to. With that being said, we all have different likes and dislikes. Unfortunately, I think in a field like this people are very impressive about what they think aren't you other people on a sense.*

(Staff, ACH 4 –Post-intervention interview)

This diversity was also related by some participants to the generation gap among the staff. Although a few participants discussed about the knowledge and expertise of new staff on personalized dementia care, five staff emphasized on sustaining the care strategies they were practicing.

*When we have a new staff member come, who apparently is young and quite well educated on the whole progressively lowered stress threshold, so you take her ideas on board and then just making sure that everyone who is here, remains on board with it. But you will nave see everyone on the same page and there are staff who do not accept new staff's ideas.*

(Staff, ACH 3 –Post-intervention focus group)

*I think it's up to all of us to make sure that and new staff members, that's right. Yeah, because when new ones come through, we still need to sustain what we're doing. So, you teach them what we've been taught along the way. . .*

(Staff, ACH 4 –Three-months follow-up interview)

*Well when it first started, we did have a couple of problems with particular staff that were, it was all new, and a lot of people* [new carers] *don't like change.*

(Staff, ACH 4 –Post-intervention interview)

Diversity in staff relating to culture and generations was identified by the participants as a drawback in building coherent relationships among the staff. The hierarchical leadership and relationships discouraged them to work as a team for incorporating the components of personalized dementia care in their everyday care service, which was also influenced by organization's physical environment and safety measures.

*Physical environment and safety.* The physical environment and safety presented a context whereby the care staff experienced physical limitations of the aged care homes such as resource scarce, staff shortage, a lack of funding and safety issues.

The participants' views in the study were that aged care homes are under-structured and understaffed for the RwD. There was an agreement on the statement of the staff that aged care homes were in a requirement of a dementia care unit, as one staff described in the following excerpt:

*We don't have a proper dementia facility here. This is our main challenge of the dementia residents are going through, that affect other residents that have dementia.* [With those looking into it] *we need a proper dementia unit because how are we going to deal with it, because we don't have a proper dementia unit, you know? So there are wandering residents and this is pretty much affects resident's stress because all the incidents happen and then, you know, how we going to prevent that.*

(Staff, ACH 5 –Post-intervention interview)

As identified in the descriptive statistics that 27% of all staff participated in this study worked full-time, and this staff shortage was described by the staff related to lack of staff and employee turnover, for example:

*Sometimes you do not have much time. They are always running because we do not have just enough staff that I think people will look at instead of long term effects of something like this they looking at short time in a sense that I need to get people ready and . . . in my shift.*

(Staff, ACH 3 –Post-intervention focus group)

*The only thing that I can see is we have a huge turnover of staff - - - so the new staff really don't know what the, the program is about.*

(Staff, ACH 2 –One-month follow-up interview)

Having no dementia care unit and a lack of full-time staff and huge employee turnover contributed to a less emphasis on the personalized dementia care to the residents, which is exacerbated by a lack of funding in residential dementia care. Some participants confirmed that they were in need of funding to establish dementia care unit and recruiting more staff to provide person-centered dementia care.

*. . . we do need more money or more funding to* [support RwD], *you know. Some people for these residents here, but half of them have dementia now. We only got 16, half of them with dementia, so there should be more support* [for RwD], *it's* [funding] *starting to be a big issue.*

(Staff, ACH 5 –One-month follow-up focus group)

*. . . It's hard to provide one on one staff for one resident because of course the funding is not available. With two staff as well for 16 residents* [each shift]. *Other people* [aged care homes] *have proper resources to deal with dementia and it will be otherwise it will just be challenging* [for us].

(Staff, ACH 5 –Post-intervention interview)

It was found in the views of the participants that this lack of staff and funding had direct impact on the maintenance of personalized dementia care, which had also influenced the safety measures for the RwD. Some staff discussed about a lack of safety for the RwD relating to residents' smoking, the fence and working alone in managing physical aggression of the RwD, for example:

*Residents do still smoke everywhere. I do not think it is not safe. We try to help them–we keep them smoking one by one and take care individually while they smoke, but most time they won't listen to us.*

(Staff, ACH 5 –Post-intervention interview)

*There is a plan for an extension of the fence which is very important as safety and security are a major priority as few residents have a tendency to walk around alone and we need more infrastructure funding.*

(Staff, ACH 5 –Three-months follow-up focus group)

*Yeah, even one resident hitting another resident with dementia, because of he followed them into her room . . . he followed her into her room and of course she doesn't like it, so she hits him, she tells him to get out from her room. We need more resources for* [managing] *things like this.*

(Staff, ACH 3 –Post-intervention interview)

In summary, the experiences and perspectives of the staff provided seven major dimensions of personalized dementia care under three spheres: a requirement of more focus on residents and family members; integrating individualized approach in care assessment and plan; a lack of staff education and training; a lack of person-centered interaction between staff and residents; negative influence of personal and social life of staff in providing care; authoritative leadership and relationships among staff; and a lack of resources and safety. There was a lack of inclusion of the voice of family members of RwD in providing care that impacted on designing the care assessment and plan. The generalized care plan was still in place because of a lack of knowledge and skills among the staff in understanding and managing challenging behaviors for every resident. A lack of on-going educational opportunity on dementia care in combination with diversity in staff, inability to cope with personalized dementia care and influence of personal life and social issues, all resulted in a sense that personalized dementia care was difficult to maintain in aged care homes. Along with staff needing more funding and resources including dementia care unit and staff, this lack contributed to a sense of absence of personalized care.

## Discussion

The study's findings indicate the personalized dementia care needs more attention in rural aged care homes in Australia, and provide insight into resident and family, staff and organization spheres that are of importance in planning and practicing person-centered care for the

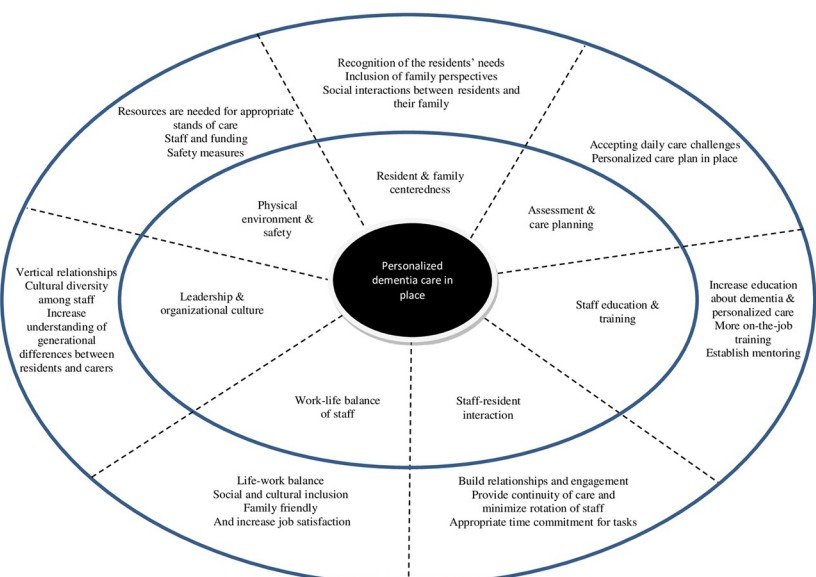

**Fig 1. A framework for integrating the quality measures into personalized care for RwD.**

RwD (Fig 1). The exploration of factors under these spheres is influenced by the multidimensional nursing home care model developed by Rantz et al. (1998), who emphasize on operational resources and components, such as central focus, interactions, milieu, environment, care, staff and safety. The factors and associated realities emerged in this study contribute to the contextualization of the dimensions based on dynamic eventualities among and between residents, staff and system. In the light of the theoretical model, acceptance and recognition of needs and inclusion of family members' perspectives in understanding the residents' needs and preferences are identified as important elements of personalized care. The meaning of personalized dementia care is shaped by the care assessment process and plan, staff competence and physical environment and safety. This study illuminates on some additional factors that shape the personalized dementia care dimensions, for example, quality of care is impacted by leadership, person-centered communication of staff with residents and personal and social life of staff. We discuss each of the dimensions and their related factors and issues as they have emerged from the study findings.

Previous studies identify a lack of recognition of the RwD and their needs and inclusion of family members' perspectives in providing care [22, 23]. Similarly, the participants in this study were unable to recognize care needs of RwD, although they reported a strong desire to support the person in a personalized manner. The way the staff determined the residents' care needs and preferences was predominantly clinical. The reasons for not being able to provide personalized care are: RwD fail to express their needs; and the perspectives of their family members are excluded [24, 25]. The exclusion of family members' voice deters the staff in translating the personalized care on the floor [26, 27]. Although the participation of family members of RwD in different programs was evident in the study, the inclusion of their voice in developing care plan and providing care was seldom seen. In fact, Moyle et al. (2011) indicates that the RwD and their family members have little control over the care management, therefore, there is a care environment that the RwD are left waiting to receive care as they want [22].

This study illustrates three factors related to personalized care assessment and plan including current care practice, personalized dementia care challenges and staff assumptions. The rural aged care homes have been attempting to integrate personalized care plan into their practice following the suggestions of Aged Care Quality Standards [28, 29]. This study findings indicate that this personalized care practice is yet to be realized because of the challenges associated with shifting from generalized to personalized care. The challenges in shifting to personalized care practice were related to myths/assumptions on behaviors of RwD and the associated care burden including workload in the translation of personalized care [30]. Studies indicate that it has been difficult for staff who provide generalized care for a long time to adapt and implement personalized care plan on the floor [30, 31]. Fear of workload increase because of a low number of permanent full-time staff [73% of staff were found work part-time or casual in this study] and personal myths/assumptions about dementia including stigma towards working with RwD contribute to avoidance to personalized care.

At the staff level, our research reflects on previous studies to indicate there is a lack of prior education about dementia and personalized care and relevant-job training opportunity. About 37% staff were from overseas who seem to have lack of knowledge of personalized dementia care as a formal medical condition. This was more evident for those staff who had a Certificate III in Aged care or disability and worked in residential aged care [14]. While the key seems to be in engaging the staff in providing personalized care to have some knowledge on dementia and therefore in the implementation of dementia care practice, they had a lack of opportunity for on-the-job training and information sessions. In essence, having a little or no knowledge of the dementia care management resulted in reduced opportunities for RwD to receive personalized care [32, 33].

The interaction of staff with RwD is an essential dimension. There is less interaction with people who are not well-expressed in aged care homes [21, 34, 35]. The need of RwD's social and intellectual interaction is poorly recognized, and staff principally focus on the residents' clinical needs because of time restrictions and task-oriented attitude [30]. As found in this study, high rotation of staff makes it difficult to build personalized interaction and meaningful relationships because of many RwD's potential difficulties in recognizing face and voice [5]. Moyle et al. (2011) suggest personalized interaction should be developed to uphold the emotional needs of RwD in connection to social well-being and intellectual stimulation that could build these residents' self-esteem and a feeling of self-worth [22]. According to Keady et al. (2007), socio-clinical biography can be a means of opening the possibility for staff to see the person beyond the dementing process and to encourage engagement in participatory activities [36].

The staff's satisfaction and dissatisfaction with the implementation of personalized dementia care are not only connected to tasks and workload, but also related with their personal and social life. Two demanding aspects of life at the same time including working with RwD and family, especially for the staff who have young family, challenge them in providing quality care service [37]. When family was inconsiderable and community did not value aged care work in this study, a feeling of isolation created that resulted in a reluctance among staff in providing personalized dementia care. Unfortunately, a feeling of isolation among staff as a result of leadership and workplace environment often goes hand in hand with the deterioration of quality in care [38].

In the organization sphere, the issues of leadership and workplace culture are of importance in the implementation of personalized care in residential dementia care. Uncoordinated dementia care management occurred when there were strained relationships among the staff with leaders and other staff due to cultural diversity and generation gaps. Authoritative leadership style discourages the care workers to implement personalized care and to be innovative in

dementia care [33]. As found in this study, Adebayo, Durey and Slack-Smith (2017) explain how cultural and linguistic differences among the staff contributed to the absence of a comprehensive oral care for RwD in aged care homes [34]. In addition, our study found the generation gap among staff affected translation of personalized care. There is a risk when cultural diversity and generation gaps create an abyss without a bridge, therefore, a breakdown in interaction and relationships is obvious between staff because the place is unknown on where to meet in the middle [32].

The accomplishment of personalized dementia care is also shaped by physical environment and safety issues that are diverse in the facilities such as infrastructure, human resources and funding. In one remote aged care home participating in this study, a potential risk of an unfenced facility was identified by participants for the RwD who were wandering. Studies investigated the capacity of aged care homes indicate a lack of infrastructure, staff and funding in rural aged care homes [39, 40]. Similar to this study findings, the staff shortage is related by Goh et al. (2017) to the employee turnover and absenteeism who note that low number of staff is the major problem in meeting the personalized care needs of RwD [41]. It is also a common belief that rural aged care homes require more funding than urban facilities to recruit and retain staff and take safety measures in order to support personalized dementia care [14, 27].

We highlight the pragmatic dimensions of personalized dementia care in the resident and family, staff and organization spheres, including central focus, care assessment and plan, staff competence, interaction between residents and staff, personal and social life of staff, leadership and relationships among staff and physical environment. This is an imperative policy insight for aged care policy makers and rural aged care homes management to consider, as it seems, in maximum cases, the current care plan and services are determined based on the staff's generalization of the needs, preference and choices of RwD. The assumption develops from the findings and discussion is that rural aged care homes requires additional efforts for a shift towards personalized dementia care.

## Limitations

The non-participation of family members of RwD in interviews and focus groups is a limitation, as is the qualitative study, perspectives of one participant group does not allow the findings to be comprehensive. However, the richness in the findings originates from the insight of a good number of staff who have been working in rural aged care homes for a long time, especially in diverse cultural and organizational contexts. The willingness of the staff in sharing their stories with RwD and the examples they provide contribute to understanding the dimensions of personalized dementia care. Another potential limitation could be social desirability bias. The participants may have answered questions in a manner that will be viewed positively by researchers. The participants were given adequate information on consent and confidentiality of the data and the opportunity to withdraw from the study at any time point. However, it may be possible the staff who had negative or alternate views did not volunteer to participate in the interview and focus group. Another limitation is that we do not explore the factors in providing personalized care in terms of the centers' size, number of residents and classification of rural locations that may have potential impact on transferability of findings. Finally, the study did not include participant observation and that the care dimensions were derived only from interviews with the participants. These are the areas that require attention in further investigation to understand the dimensions and nature of factors involved in personalized dementia care implementation, particularly in rural facilities.

## Conclusion

The dimensions associated with the quality of personalized dementia care in rural aged care homes emerged from the data provided by care staff in five rural aged care homes in Queensland and South Australia, Australia. Personalized dementia care is multidimensional and can be described using a conceptual framework that incorporates dialectical relationships and events occur in aged care homes. To achieve quality care, the dimensions that are of importance for the aged care homes include resident and family centeredness, assessment and care planning, staff education and training, staff-resident interaction, work-life balance, leadership and organizational culture and physical environment and safety.

The current service delivery focusses too much on perceived generalized clinical care needs, rather than specific person-centered care. The inclusion of the voice of residents and their family members in designing care plan is in need to establish personalized care practice for RwD to give them opportunity to live with a control over care service. Emphasis on the staff retention and recruitment and resource adequacy is important to improve quality of personalized dementia care in rural aged care homes. The aged care policy makers and organization leaders should consider work-life balance of staff as a key issue and respond with mentorship and formal support structures.

The study's findings will contribute to the knowledge of dimensions of personalized dementia care. It is evident that rural aged care homes require additional commitment from the policy makers relating to resource stabilization and improvement. This study confirms the need for a holistic approach to respond to the barriers and integrate personalized care to improve the quality of life of RwD in rural aged care homes.

## Supporting information

**S1 File. Interview group.** Interview questions for aged care manager/staff.
(DOCX)

**S2 File. Focus group schedule.** Questions and discussion points.
(DOCX)

## Acknowledgments

We are thankful to the residents, staff, and facility managers who provided time and shared their experiences about residents with dementia. We thank nursing consultants and geriatrician who assisted in the implementation of the intervention program. Acknowledgement also goes to research assistant at Flinders University Rural Health SA, for her valuable support in data collection.

## Author Contributions

**Conceptualization:** Mohammad Hamiduzzaman, Abraham Kuot, Vivian Isaac.

**Data curation:** Mohammad Hamiduzzaman, Abraham Kuot.

**Formal analysis:** Mohammad Hamiduzzaman, Abraham Kuot, Jennene Greenhill.

**Funding acquisition:** Mohammad Hamiduzzaman, Vivian Isaac.

**Investigation:** Mohammad Hamiduzzaman, Vivian Isaac.

**Methodology:** Mohammad Hamiduzzaman, Edward Strivens, Vivian Isaac.

**Project administration:** Mohammad Hamiduzzaman, Abraham Kuot.

**Resources:** Mohammad Hamiduzzaman.

**Software:** Mohammad Hamiduzzaman.

**Supervision:** Jennene Greenhill, Vivian Isaac.

**Validation:** Mohammad Hamiduzzaman, Jennene Greenhill, Edward Strivens.

**Writing – original draft:** Mohammad Hamiduzzaman.

**Writing – review & editing:** Mohammad Hamiduzzaman, Abraham Kuot, Jennene Greenhill, Edward Strivens, Vivian Isaac.

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
