## [Decision Letter · Decision Letter 0]

25 Mar 2020

PONE-D-20-03933

Towards Personalized Care: Factors Associated with the Quality of Life of Residents with Dementia in Australian Rural Aged Care Homes

PLOS ONE

Dear Dr Hamiduzzaman,

Thank you for submitting your manuscript to PLOS ONE. After careful consideration, we feel that it has merit but does not fully meet PLOS ONE’s publication criteria as it currently stands. Therefore, we invite you to submit a revised version of the manuscript that addresses the points raised during the review process.

We would appreciate receiving your revised manuscript by May 09 2020 11:59PM. To enhance the reproducibility of your results, we recommend that if applicable you deposit your laboratory protocols in protocols.io, where a protocol can be assigned its own identifier (DOI) such that it can be cited independently in the future. For instructions see: http://journals.plos.org/plosone/s/submission-guidelines#loc-laboratory-protocols

We look forward to receiving your revised manuscript.

Kind regards,

Tim Luckett

Academic Editor

PLOS ONE

Journal Requirements:

2. Please ensure you have thoroughly discussed any potential limitations of this study within the Discussion section. In addition, please include additional information regarding the interview guide used in the study and ensure that you have provided sufficient details that others could replicate the analyses. For instance, if you developed a guide as part of this study and it is not under a copyright more restrictive than CC-BY, please include a copy, in both the original language and English, as Supporting Information.

Reviewers' comments:

Reviewer's Responses to Questions

**Comments to the Author**

1. Is the manuscript technically sound, and do the data support the conclusions?

Reviewer #1: Partly

Reviewer #2: Partly

2. Has the statistical analysis been performed appropriately and rigorously? 

Reviewer #1: N/A

Reviewer #2: I Don't Know

3. Have the authors made all data underlying the findings in their manuscript fully available?

Reviewer #1: Yes

Reviewer #2: Yes

4. Is the manuscript presented in an intelligible fashion and written in standard English?

Reviewer #1: Yes

Reviewer #2: Yes

5. Review Comments to the Author

Reviewer #1: Thank you for the opportunity to review this manuscript exploring factors associated with the quality of life of residents with dementia in Australian rural aged care homes. This is a valuable topic and the need for research in this area and aims of this study are well-defined. I do have some concerns about the presentation of results, which I feel could be more clearly articulated and linked back to the theoretical framework noted in the methods section. I have listed more specific points below for your consideration that I hope will be helpful.

• Aims presented clearly in the abstract and it is evident to readers that this paper reports qual findings of the larger mixed-methods study.

• Further information about recruitment is needed – it is noted that managers of each site were contacted to recruit the aged care workers, but what measures were taken to avoid potential feelings of coercion? How were the workers invited to participate in the study and how was the information sheet and consent form provided to potential participants? Was this in person? Email? And by whom? More detail needed.

• Please check the English and sentence structure throughout the manuscript. e.g. ‘Some family members were labelled…’ or ‘…intervenes a non-pharmacological personalised dementia care model...’ - what does this mean?

• It is very unclear how many participants were involved in this qualitative portion of the study. Participant characteristics were presented for the entire cohort which may or may not reflect the characteristics of the qualitative sub-sample. It would be good to present demographic information about the interviewees/focus group participants only so we know whose perspectives are being represented in this paper.

• A major concern is that quotes don’t seem to support what is being said in the text in every instance. E.g. do the quotes on pg. 7 really illustrate ‘excluding’ family member’s perspectives? Please ensure quotes are supporting of themes or that further explanation in-text is included to provide clarity about how quotes relates to the theme.

• All quotes can be condensed to enhance readability.

• Interviews and focus groups were conducted at different time points…does this add to these findings in any way?

• Please write out acronyms used in the first instance in full (quote on pg. 11)

• Please ensure font size for all text in the figures is consistent – ‘Personalised care plan in place’

• Given the theoretical background presented in the methods section, I expected to see further discussion around how the results fit/did not fit with the theoretical framework.

Reviewer #2: This study “Harmony in the Bush” explored the factors that shape the dimensions of personalized dementia care in rural nursing homes using data from a large dementia study through a quasi-experimental and mixed methods approach (appears to be a secondary data analysis) Harmony in the Bush is a personalized dementia care model.

Abstract methods:

1) no description of quasi-experimental approach. What was quasi-experimental in this study?

2) You state in the background that the aim to explore the factors that shape the dimensions of personalized dementia care in rural nursing homes using data from a large dementia study. In the conclusion you state Understanding the dimensions of quality of care is important to achieve an effective personalized approach in dementia care. This is confusing and does not reflect the results. You stated in the introduction that you want to understand the dimensions of personalized care to inform an approach that is of high quality but it is mixed up in the conclusion and makes the study purpose and conclusion disjointed.

Introduction:

1) First line is confusing. Should it state: personalized care is integral to the quality of dementia care”? because later in the paragraph you state: “The literature indicates a personalized approach, based on active listening, recognition, compassion, attentiveness and sensitivity to the resident’s needs and preferences in providing care, promotes quality of life for residents with dementia”

2) What previous work has been done on “dimensions of personalized aged care”?

Theoretical Background:

1) Consider making a figure or table of the text describing the multidimensional theoretical model developed by Rantz et al. to better present this model. The text is too dense and hard to follow

Materials and Methods:

1) Again, where is the quasi-experimental approach and description?

2) If this is a secondary analysis of qualitative data please state this and how the data was appropriate for your research question/aim if this study. You state in the intro that data are from the Harmony in the Bush project then below that state the data are part of a larger project. It is not clear if the study is designed to answer your question or is you decided post-hoc to ask this question. This does have implications for the design and method being appropriate for the question and you will need to add justification as to why this approach is appropriate.

Data Collection:

1) Vert good attention to ethics approval and staff protection!

2) How/when did you know you have collected enough data to come to the themes that emerged?

Data analysis:

1) confusing when multiple terms and processes related to the analysis (automatic search, codes, nodes, topic coding, derivative categories, provisional concept map, themes, sub-themes, categories) are used but not shown how one analytic step informed another. Consider using consistent terms and/or clearly defining how each of these analytic steps led to the next. As of now it is not clear and that might be because you are using so many undefined terms. It makes the reader a little hesitant to believe the findings because it is not clear how you got from data collection to findings.

2) You do a nice job attending to potential bias but you might use a reference to support the approach “maintain reliability and validity of the findings” one is: Creswell J. Chapter 10: Standards of quality and verification. Qualitative Inquiry and Research Design: Choosing Among Five Traditions. Thousand Oaks, CA: Sage; 1998:193-218.

Findings

1) Born in overseas- do you have an idea of where? Even if you can break this down by continent since it is close to half of the sample.

2) Wow- 37% oversees and 73% part-time and casual staff, How does this impact your findings? Maybe touch on in the discussion.

3) The findings seem to fall into three larger themes around issues with: family/resident, staff, and organization.

Theme: Family/resident, Subthemes: assessment/care planning, resident and family centeredness

Theme: Staff, Subthemes: work life balance, staff resident interaction, staff education/training

Theme: Organization, Subthemes: physical environment, leadership and organizational culture

This makes me hesitant to believe that data analysis was fully completed and there may be another step in the analysis.

Discussion

Figure 1 is excellent and I think with additional analysis it can provide a useful model for designing interventions to improve person centered-ness dementia care.

6. PLOS authors have the option to publish the peer review history of their article (what does this mean?). If published, this will include your full peer review and any attached files.

Reviewer #1: No

Reviewer #2: No

---

## [Author Response · Author response to Decision Letter 0]

14 Apr 2020

To

Tim Luckett, Academic Editor

PLOS ONE

RE: Revised manuscript “Towards personalized care: Factors associated with the quality of life of residents with dementia in Australian rural aged care homes” [PONE-D-20-03933]

Dear Dr Lackett,

Thank you for the opportunity to strengthen our manuscript submission. We found the reviewer’s comments helpful and have addressed them in the point-by-point table below.

Please do not hesitate to contact me should you have any queries to the revision.

We thank you for your time and look forward to hearing from PLOS ONE in regard to our manuscript.

A point-by-point response letter has been accompanied with our revised manuscript with track changes and final manuscript. This letter provides a detailed response to each reviewer/editorial point raised, describing exactly what amendments have been made to the manuscript text and where these can be viewed. All changes to the manuscript are indicated in the text by highlighting or using track changes [See in Attached Response Letter].

Kind Regards

Dr Mohammad Hamiduzzaman

Research Fellow

Flinders University Rural Health SA

College of Medicine & Public Health

Flinders University

---

## [Decision Letter · Decision Letter 1]

6 May 2020

Towards personalized care: Factors associated with the quality of life of residents with dementia in Australian rural aged care homes

PONE-D-20-03933R1

Dear Dr. Hamiduzzaman,

We are pleased to inform you that your manuscript has been judged scientifically suitable for publication and will be formally accepted for publication once it complies with all outstanding technical requirements.

With kind regards,

Tim Luckett

Academic Editor

PLOS ONE

Reviewer's Responses to Questions

**Comments to the Author**

1. If the authors have adequately addressed your comments raised in a previous round of review and you feel that this manuscript is now acceptable for publication, you may indicate that here to bypass the “Comments to the Author” section, enter your conflict of interest statement in the “Confidential to Editor” section, and submit your "Accept" recommendation.

Reviewer #2: All comments have been addressed

2. Is the manuscript technically sound, and do the data support the conclusions?

Reviewer #2: Yes

3. Has the statistical analysis been performed appropriately and rigorously? 

Reviewer #2: N/A

4. Have the authors made all data underlying the findings in their manuscript fully available?

Reviewer #2: Yes

5. Is the manuscript presented in an intelligible fashion and written in standard English?

Reviewer #2: Yes

6. Review Comments to the Author

Reviewer #2: All points addressed and I have no concerns about the revised manuscript. This is a valuable contribution to dementia care.

7. PLOS authors have the option to publish the peer review history of their article (what does this mean?). If published, this will include your full peer review and any attached files.

Reviewer #2: No

---

## [Editor Report · Acceptance letter]

11 May 2020

PONE-D-20-03933R1 

Towards personalized care: Factors associated with the quality of life of residents with dementia in Australian rural aged care homes 

Dear Dr. Hamiduzzaman:

I am pleased to inform you that your manuscript has been deemed suitable for publication in PLOS ONE. Congratulations! Your manuscript is now with our production department. 

With kind regards,

on behalf of

Dr. Tim Luckett 

Academic Editor

PLOS ONE